# Epithelial Cell Line Derived from Endometriotic Lesion Mimics Macrophage Nervous Mechanism of Pain Generation on Proteome and Metabolome Levels

**DOI:** 10.3390/biom11081230

**Published:** 2021-08-17

**Authors:** Benjamin Neuditschko, Marlene Leibetseder, Julia Brunmair, Gerhard Hagn, Lukas Skos, Marlene C. Gerner, Samuel M. Meier-Menches, Iveta Yotova, Christopher Gerner

**Affiliations:** 1Department of Analytical Chemistry, Faculty of Chemistry, University of Vienna, Waehringer Straße 38, 1090 Vienna, Austria; benjamin.neuditschko@univie.ac.at (B.N.); marlene_leibetseder@gmx.at (M.L.); julia.brunmair@univie.ac.at (J.B.); gerhard.hagn@univie.ac.at (G.H.); lukas.skos@univie.ac.at (L.S.); samuel.meier@univie.ac.at (S.M.M.-M.); 2Department of Inorganic Chemistry, Faculty of Chemistry, University of Vienna, Waehringer Straße 42, 1090 Vienna, Austria; 3Division of Biomedical Science, University of Applied Sciences, FH Campus Wien, Favoritenstraße 226, 1100 Vienna, Austria; marlene.gerner@fh-campuswien.ac.at; 4Joint Metabolome Facility, Faculty of Chemistry, University of Vienna, Waehringer Straße 38, 1090 Vienna, Austria; 5Department of Obstetrics and Gynaecology, Medical University of Vienna, Spitalgasse 23, 1090 Vienna, Austria; iveta.yotova@meduniwien.ac.at

**Keywords:** endometriosis, inflammation, metabolomics, multi-omics, proteomics

## Abstract

Endometriosis is a benign disease affecting one in ten women of reproductive age worldwide. Although the pain level is not correlated to the extent of the disease, it is still one of the cardinal symptoms strongly affecting the patients’ quality of life. Yet, a molecular mechanism of this pathology, including the formation of pain, remains to be defined. Recent studies have indicated a close interaction between newly generated nerve cells and macrophages, leading to neurogenic inflammation in the pelvic area. In this context, the responsiveness of an endometriotic cell culture model was characterized upon inflammatory stimulation by employing a multi-omics approach, including proteomics, metabolomics and eicosanoid analysis. Differential proteomic profiling of the 12-Z endometriotic cell line treated with TNFα and IL1β unexpectedly showed that the inflammatory stimulation was able to induce a protein signature associated with neuroangiogenesis, specifically including neuropilins (NRP1/2). Untargeted metabolomic profiling in the same setup further revealed that the endometriotic cells were capable of the autonomous production of 7,8-dihydrobiopterin (BH2), 7,8-dihydroneopterin, normetanephrine and epinephrine. These metabolites are related to the development of neuropathic pain and the former three were found up-regulated upon inflammatory stimulation. Additionally, 12-Z cells were found to secrete the mono-oxygenated oxylipin 16-HETE, a known inhibitor of neutrophil aggregation and adhesion. Thus, inflammatory stimulation of endometriotic 12-Z cells led to specific protein and metabolite expression changes suggesting a direct involvement of these epithelial-like cells in endometriosis pain development.

## 1. Introduction

Endometriosis is a chronic inflammatory disease describing the abnormal growth of uterine tissue outside of the uterine cavity in the pelvic area [1]. Endometriotic cells are characterized by invasive phenotypes. They successfully attach to pelvic organs and cause pelvic inflammation [2,3]. Studies estimate that the disease is affecting about 1 in 10 women worldwide. The clinical symptoms include dysmenorrhea, dyspareunia, chronic pelvic pain and infertility. To date, there exists no curative treatment [1]. As one of the cardinal symptoms, pain strongly affects the patients’ quality of life and can only be treated symptomatically so far. A large fraction of the published studies focused on phenotypic investigations and the molecular mechanisms, especially those associated with pain development, remain to be fully elucidated [4].

According to the rAFS/ASRM system, the extent of the disease does not correlate with the pain level experienced by individual patients [5]. The severity of the pain sensation seems to be connected to a mixture of neuropathy, neurogenic inflammation, nociception and hyperalgesia [6]. It is known that the lesion’s surrounding nerves are infiltrated and compressed by endometriotic cells [7]. Moreover, there is evidence that nascent nerve cells can be attracted by the endometriotic lesion. Their inflammatory activated state may also directly cause and transmit pain stimuli to the central nervous system [8]. Some nerve cells may even release proinflammatory factors, which ultimately lead to neurogenic inflammation and increased local vascular permeability, enhancing migration of ectopic endometrial cells [9,10,11]. Furthermore, it is well accepted that endometriosis-associated pain is directly linked to dorsal root ganglion neurons [12].

Immunological dysregulation seems to represent a main pathogenesis driver in endometriosis [13]. Normally, cell-mediated immune responses contribute to the elimination of immune invaders and clearance of ectopic endometrial tissue. In endometriosis, the clearance of endometrial tissue in the peritoneal cavity is abolished due to an impaired immune response at the site of implantation [14,15]. Deregulated T-cell immunity and a suppressed activity of macrophages and NK cells were found to contribute to this process [16]. The activation of an inflammatory response in the peritoneum of women with the disease leads to local production of cytokines, chemokines and growth factors that enhance the growth of the ectopic endometrial tissue by inhibiting normal apoptotic processes and promoting local angiogenesis and neurogenesis [17]. The macrophage-nervous axis in endometriosis is commonly accepted to be the main cause for disease-associated pain [18]. Indeed, nerve infiltration is positively correlated with high density of tissue-resident macrophages in the lesion [19]. Alternatively, these nerve fibres are attracted by the action of semaphorins. As semaphorins normally regulate axon migration, axonal growth and guidance, altered semaphorin secretion may lead to aberrant nervous innervation in endometriosis [18]. Infiltrating the endometriotic lesions, they secrete neuroangiogenic factors and create a neovasculative environment [20]. Herein, especially VEGF secretion plays an important role in axonal outgrowth functioning as a neurotrophic factor [21,22]. Once neuroangiogenesis and infiltration in nerve fibres was initiated by aberrant inflammatory signaling, the endometriotic lesion may be create its own altered microenvironment [23].

Macrophages secrete tumour necrosis factor α (TNFα) and interleukin-1β (IL1β) that contribute to disease progression [24,25]. These inflammatory cytokines can mediate neurogenic inflammation and secretion of further neuroangiogenic factors [26,27]. TNFα signaling increases the transient receptor potential vanilloid 1 (TRPV1) nociception in the dorsal root neurons which contributes to hyperalgesia and neuropathic pain sensation [28,29]. It was also found to sensitize sensory nerves to a constant induction of the action potential via TRPV1 in patients, mainly through overexpression of voltage-gated sodium channels [27]. TRPV1 expression is also increased in uterosacral ligaments in endometriosis patients [30] and it has been shown that pain is often driven by dorsal root ganglion neurons, in association with TRPV1 [12]. Neurogenesis seems to be at least partially responsible for neuropathic pain experiences [31,32]. Therefore, the molecular mechanisms of neurogenesis in endometriosis need to be comprehensively characterized to contribute towards novel therapy options in endometriosis pain management.

During the last years proteomics and metabolomics analysis were applied to uncover markers for early detection of endometriosis and to understand the molecular changes associated with its pathophysiology [33,34,35]. However, the use of these omics techniques is still sparse in endometriosis and multi-omics profiling was not yet applied, to the best of our knowledge. Combining proteomic with metabolomic profiling is especially attractive, since it may support a functional interpretation of the involved pathways. In addition, signaling lipids are key players during inflamation and act in a concerted fashion with proteins [36,37]. Thus, we have applied an in-depth proteome, metabolome and eicosanoid profiling of the endometriotic epithelial cell line 12-Z to investigate and characterize their responsiveness towards inflammatory signals. The cell line was isolated from a patient with peritoneal endometriosis and immortalized by Starzinski-Powitz [38]. 12-Z was characterized as epithelial-like (cytokeratin-positive/E-cadherin negative) and using a matrigel assay it was shown that the cell line was highly invasive [38]. To simulate the inflammatory macrophage signaling, the cells were treated with the cytokines TNFα or IL1β, which are typically upregulated in endometriotic lesions of patients [39]. We provide proteomic and metabolic evidence that the endometriotic 12-Z cells may independently express key mediators of neuroangiogenesis and neuropathic pain.

## 2. Material & Methods

### 2.1. Cell Culture

The 12-Z cell line was a generous gift of Dr. Anna Starzinski-Powitz (Goethe-Universität Frankfurt) [38]. Human epithelial endometriotic cell line 12-Z was cultivated in DMEM-F12 phenolredfree (Gibco, Thermo Fisher Scientific, Vienna, Austria) with 10% (*v/v*) heat inactivated fetal calf serum (FCS, SigmaAldrich, Vienna, Austria) and 1% (*v/v*) penicillin and streptomycin (Sigma-Aldrich, Vienna, Austria). Cultivation was done in humidified incubators at 37 °C and 5% CO_2_. The 12-Z cell line was grown in T75 polystyrene cell culture flasks with cell growth surface for adherent cells (Sarstedt, Wiener Neudorf, Austria). Cells were sub-cultured every 3–4 days at a 1:3 ratio. Cells were cultivated until they reached a concentration of 80% confluency. Cell counting was performed using a MOXI Z mini automated cell counter (ORFLO Technologies, Ketchum, ID, USA) using Moxi Z Type M cassettes (ORFLO Technologies, Ketchum, ID, USA). Cells were routinely checked for mycoplasma contamination using MycoAlert™ mycoplasma detection kit (Szabo-Scanidc, Vienna, Austria). Cells were seeded on a 6-well plate with cell growth surface for adherent cells (Sarstedt, Wiener Neudorf, Austria) at a density of 300.000 cells/well in 3 mL growth medium. Inflammatory stimulation was applied for 48 h at a concentration of 10 ng mL^−1^ with either TNFα or IL1β (both Sigma-Aldrich).

### 2.2. Proteomics

After the indicated treatment, the cells were washed twice with PBS and fractionation was performed as previously described [40]. Isotonic lysis buffer (1 mL of 10mM HEPES/NaOH, pH 7.4, 0.25 M sucrose, 10 mM NaCl, 3 mM MgCl_2_, 0.5% Triton × 100, protease and phosphatase inhibitor cocktail (Sigma-Aldrich)) was added to the cells, which were scraped using a cell scraper. Cell lysis was achieved using mechanical shear stress employing a syringe and a needle. After centrifugation at 2270× *g* for 5 min supernatant containing cytoplasmic proteins was precipitated in 4 mL of cold EtOH overnight.

Protein sample preparation: The ethanolic protein suspension was centrifuged at 4536× *g* for 30 min (4 °C). The supernatant was discarded, while the protein pellet was dried and resuspended in lysis buffer (8 M urea, 50 mM TEAB, 5% SDS). Protein concentration was determined using the bicinchoninic acid assay. An aliquot of the samples containing 20 µg protein was digested using a modified Protifi protocol [41]. In short, the samples were diluted to a concentration of 1 µg µL^−1^. The diluted sample (20 µg in 20 µL) were pipetted into a 1.5 mL Eppendorf microcentrifuge tube and 20 µL of DTT (64 mM) was added. The samples were heated for 10 min at 95 °C under constant shaking (300 rpm). The samples were cooled to room temperature, treated with iodacetamide (5 µL of 486 mM solution, and incubated in the dark for 30 min at 30 °C and 300 rpm. Afterwards, phosphoric acid (4.5 µL of 12%) was added, resulting in 1.2% final concentration of phosphoric acid. S-Trap buffer (297 µL, 90% MeOH (*v/v*), 0.1 M TEAB) was added to the solution. The sample was loaded on the Protifi S-Trap column and washed 4 times with S-trap buffer (150 µL). Trypsin/LysC (MS grade; Promega Corporation, Madison, WI, USA) was added in a 1:40 ratio (0.5 µg for 20 µg protein). After digestion for 2 h at 37 °C peptides were eluted, dried using a SpeedVac and stored until further analysis.

Data acquisition: Dried peptides were reconstituted in 5 µL 30% formic acid, containing four synthetic peptides, for quality control [42]. The samples were further diluted with 40 µL mobile phase A (97.9% H_2_O, 2% acetonitrile, 0.1% formic acid). Peptides were analyzed with a Dionex UltiMate 3000 Nano LC system coupled to a Q Exactive Orbitrap mass spectrometer (Thermo Fisher Scientific, Vienna, Austria) using a previously published method [40]. In short, peptides were separated on a 50 cm × 75 μm PepMap100 analytical column (Thermo Fisher Scientific), at a flow rate of 300 nL min^−1^. Gradient elution of the peptides was achieved by increasing the mobile phase B (79.9% acetonitrile, 20% H_2_O, 0.1% formic acid) from 8% to 40%, with a total chromatographic run time of 135 min including washing and equilibration. Mass spectrometric resolution on the MS1 level was set to 70,000 (at *m/z* 200) with a scan range from *m/z* 400–1400. The 12 most abundant peptide ions were selected for fragmentation (Top12) at 30% normalized collision energy and analyzed in the Orbitrap at a resolution of 17,500 (at *m/z* 200). 

Data analysis: Data was analyzedanalyzed in settings as previously described [42]. Briefly, raw data was submitted to the freely available software MaxQuant (version 1.6.6.0) [43] utilizing the Andromeda search engine. A minimum of two peptide identifications, at least one of them being a unique peptide, was required for valid protein identification. The false discovery rate (FDR) was set to 0.01 on both peptide and protein level. Uniprot database (Human, version 03/2018, reviewed entries only, 20,316 protein entries) was used to generate the fasta file used for the search. For statistical data evaluation MaxQuant companion software Perseus (version 1.6.1.0) was used. Reverse sequences and potential contaminants as well as proteins identified only by site were removed. Label-free quantification (LFQ) values were converted to Log2(x) and technical replicates averaged. Proteins were filtered for valid values, keeping only proteins that were identified in at least three measurements of one sample group. Evaluation of regulatory events between different samples groups was achieved by two-sided t-tests using a FDR < 0.05 calculated by permutation-based test [44,45]. Significantly regulated proteins were further analyzed using the Cytoscape [46] plugin ClueGo [47] with default settings. Gene ontology for Biological Processes (GOBP) was used as search space with medium network specificity. Statistical options were set to two-sided hypergeometriy test with Bonferroni step down p-value correction. String [48] analysis was further used to display protein network connections of regulated proteins. STRING protein query of species homo sapiens was used with a confidence (score) cut-off of 0.4 and with 0 additional interactors allowed. The mass spectrometry proteomics data have been deposited to the ProteomeXchange Consortium via the PRIDE [49] partner repository with the dataset identifier PXD022354 and 10.6019/PXD022354. 

### 2.3. Eicosanoid Analysis

Eicosanoid sample collection and preparation: Supernatants from cell culture experiments or medium (3 mL) was added to cold ethanol (12 mL, EtOH, abs. 99%, −20 °C; AustroAlco) containing an internal standard mixture of 12S-HETE-d8, 15S-Hete-d8, 5-Oxo-ETE-d7, 11.12-DiHETrE-d11, PGE-d4 and 20-HETE-d6 (each 100 nM; Cayman Europe, Tallinn, Estonia). The samples were stored over-night at −20 °C. After centrifugation (30 min, 5000 rpm, 4 °C), the supernatant was transferred to a new 15 mL Falcon™ tube and EtOH was evaporated via vacuum centrifugation (37 °C) until the original sample volume was accomplished. Samples were loaded on preconditioned StrataX solid phase extraction (SPE) columns (30 mg mL^−1^; Phenomenex, Torrance, CA, USA) using Pasteur pipettes. SPE columns were washed with MS grade water (3 mL) and elution of eicosanoids was performed with ice-cold methanol (500 µL, MeOH abs.; VWR International, Vienna, Austria) containing 2% formic acid (FA; Sigma-Aldrich). MeOH was evaporated under a gentle stream of nitrogen at room temperature and the samples were reconstituted in 150 µL reconstitution buffer (H_2_O:ACN:MeOH + 0.2% FA–vol% 65:31.5:3.5), including a second mixture of internal standards, including 5S-HETE-d8, 14.15-DiHETrE-d11 and 8-iso-PGF2a-d4 (10–100 nM; Cayman Europe, Tallinn, Estonia). Reconstituted samples were stored at +4 °C and measured subsequently via LC-MS/MS.

Data acquisition: Separation of eicosanoids was performed on a Thermo Scientific™ Vanquish™ (UHPLC) system equipped with a Kinetex^®^ C18-column (2.6 μm C18 100 Å, LC Column 150 × 2.1 mm; Phenomenex^®^). All samples were analyzed in technical duplicates. The injection volume was 20 µL and the flow rate was kept at 200 µL min^−1^. The UHPLC method included a gradient flow profile (mobile phase A: H_2_O + 0.2% FA, mobile phase B: ACN:MeOH (vol% 90:10) + 0.2% FA) starting at 35% B and increasing to 90% B (1–10 min), further increasing to 99% B within 0.5 min and held for 5 min. Afterwards solvent B was decreased to the initial level of 35% within 0.5 min and the column was equilibrated for 4 min, resulting in a total run time of 20 min. The column oven temperature was set to 40 °C. The UHPLC system was coupled to a Q Exactive™ HF Quadrupole-Orbitrap™ mass spectrometer (Thermo Fisher Scientific, Austria), equipped with a HESI source for negative ionization to perform the mass spectrometric analysis. The resolution on the MS1 level was set to 60,000 (at *m/z* 200) with a scan range from *m/z* 250–700. The two most abundant precursor ions were picked for fragmentation (HCD 24 normalized collision energy), preferentially from an inclusion list containing *m/z* 31 values specific for eicosanoids and their precursor molecules. Resulting fragments were analyzed on the MS2 level at a resolution of 15,000 (at *m/z* 200). Operating in negative ionization mode, a spray voltage of 2.2 kV and a capillary temperature of 253 °C were applied. Sheath gas was set to 46 and the auxiliary gas to 10 (arbitrary units).

Data analysis: Data interpretation of raw files generated by the Q Exactive™ HF Quadrupole-Orbitrap™ mass spectrometer was performed manually using Thermo Xcalibur™ 4.1.31.9 (Qual browser). Spectra were compared with reference spectra from the Lipid Maps depository library from July 2018 [50]. Peaks were integrated using the TraceFinder™ software package (version 4.1—Thermo Scientific, Vienna, Austria).

### 2.4. Metabolomics

Metabolomic sample collection and preparation: For metabolomics analysis, cells were seeded at 10^6^ cells per T25 flask in complete medium (5 mL) and left to adhere over-night. They were then treated with IL1β or TNFα for 48 h similarly to the proteomic experiment. Thereafter, the medium was removed and centrifuged (1100 rpm, 2 min, 4 °C). An aliquot (200 µL) of each medium sample was precipitated in cold MeOH (100%, 800 µL) and stored at −80 °C. The methanolic solution contained dopamine-d4, melatonin-d4 (both Santa Cruz Biotechnology, Dallas, TX, USA) and *N*-acetyl-serotonin-d3 (Toronto Research Chemicals BIOZOL) as internal standards at concentrations of 120 pg µL^−1^. The cell samples were washed once with PBS (3 mL) and metabolites were extracted with cold MeOH (80%, containing stds, 1 mL). The 80% methanolic solution contained dopamine-d4, melatonin-d4 (both Santa Cruz Biotechnology, Dallas, TX, USA) and *N*-acetyl-serotonin-d3 (Toronto Research Chemicals BIOZOL) as internal standards at concentrations of 100 pg µL^−1^. Each flask was processed at a time and immediately snap-frozen in liquid nitrogen. Three replicates per condition were then thawed together and the cells were scraped, transferred into labelled Eppendorf tubes and were stored at −80 °C. Samples were dried and reconstituted in 120 µL of 1% methanol and 0.2% formic acid and 1 pg µL^−1^ caffeine-(trimethyl-D9) (Sigma Aldrich) and transferred into HPLC vials equipped with a 200 µL V-shape glass insert (both Macherey-Nagel GmbH Co. KG) suitable for LC-MS/MS analysis. Caffeine-(trimethyl-d9) (1 pg µL^−1^) was used as an additional internal standard. Again, the experiment was carried out in biological triplicates.

Data acquisition: Samples were separated on a reversed phase Kinetex XB-C18 column (2.6 µm, 150 × 2.1 mm, 100 Å, Phenomenex Inc., Torrance, CA, USA) using a Vanquish UHPLC System (Thermo Fisher Scientific). Mass spectrometric analysis was performed on a Q Exactive HF orbitrap (Thermo Fisher Scientific). Mobile phase A consisted of water with 0.2% formic acid, mobile phase B of methanol with 0.2% formic acid and the following gradient program was run: 1 to 5% B in 0.5 min, 5 to 40% B from 0.5–5 min, then 40 to 90% B from 5–8 min, followed by a wash phase at 90% B for 2.5 min and then an equilibration phase at 1% B for 2 min, yielding a total run-time of 12.5 min. Flow rate was 0.5 mL min^−1^, injection volume was 5 µL and the column temperature was set to 40 °C. The injection needle was washed in between runs with 10% methanol. All samples were analyzed in technical replicates. Samples were analyzed in positive, as well as in negative ionization mode. Scan range was from *m/z* 100–1000 and resolution was set to 60,000 (at *m/z* 200) for MS1 and 15,000 (at *m/z* 200) for MS2. The four most abundant ions of the full scan were selected for further fragmentation in the HCD collision cell applying 30 eV normalized collision energy. Dynamic exclusion was applied for 6 s. Instrument control was performed using Xcalibur 4.0 Qual browser (Thermo Fisher Scientific).

Data analysis: Raw files generated by the Q Exactive HF were loaded into the Compound Discoverer Software 3.1 (Thermo Fisher Scientific). Compounds were identified in Compound Discoverer with a user workflow tree. A maximum retention time shift of 0.1 min was allowed for aligning features and using a maximal mass tolerance of 5 ppm. Metabolites were matched against mzcloud (Copyright © 2021–2020 HighChem LLC, Slovakia mzCloud is a trademark of HighChem LLC, Bratislava, Slovakia). Compounds with a match factor ≥80 were manually checked. This was performed with Xcalibur 4.0 Qual browser (ThermoFisher Scientific). For peak integration and calculation of peak areas, the Tracefinder Software 4.1 (ThermoFisher Scientific) was used. The generated batch table was exported and further processed with Microsoft Excel, GraphPad Prism (Version 6.07) and the Perseus software (Version 1.6.12).

### 2.5. Cell Cycle Analysis

Flow cytometry was performed to determine the cell cycle distribution with and without inflammatory stimulation. Therefore, BD Cycletest™ Plus DNA Kit (BD Biosciences, Vienna, Austria) was used according to the manufacturer’s protocol to prepare the cells and measured on a CytoFLEX Flow Cytometer (Beckman&Coulter, Vienna, Austria) in the PE-channel. Statistical significance was evaluated with a bidirectional student t-test and three biological replicates.

## 3. Results

Macrophages can stimulate endometriotic cells by secreting cytokines such as TNFα and IL1β [24,25]. Here, an in-vitro model of endometriosis was investigated to evaluate the effects of such inflammatory stimuli on endometriotic cells by means of a multi-omics approach, including untargeted shotgun proteomics, metabolomics and eicosanoid analysis (Appendix A). The endometriotic 12-Z cell line was treated with TNFα or IL1β at 10 ng mL^−1^ for 48 h (Figure 1A). Flow cytometry analysis showed a slight increase in the tetraploid G2/M phase, as well as S-Phase and a corresponding decrease of cells in G0/G1 phase (Appendix A).

### 3.1. Eicosanoid Analysis Reveals the Uptake of Eicosanoid Precursors by 12-Z Cells from the Growth Medium

Eicosanoids from the supernatants of control and inflammatory stimulated 12-Z cells were enriched by a solid-phase extraction protocol and subsequently analyzed by mass spectrometry. A total of 49 eicosanoids and polyunsaturated fatty acids (PUFAs) were detected in the supernatants. The composition of the fully supplemented medium was additionally verified. The epithelial-like 12-Z cells efficiently depleted the growth medium of the eicosanoid precursors arachidonic acid (AA), docosahexaenoic acid (DHA) and eicosapentaenoic acid (EPA) irrespective of treatment condition (Appendix A). Strikingly, EPA was not detectable after culturing the cells for 48 h. In contrast, the mono-oxygenated hydroxyeicosatetraenoic acids 16-HETE and 18-HETE were not detected in the growth medium, but only in the presence of the cultured cells. Inflammatory stimulation had little impact on the differential expression of eicosanoids.

### 3.2. TNFα- and IL1β-Stimulated 12-Z Cells Show Enhanced Proliferation and Activation of IL-12 and NF-κB Signaling Pathways

Proteomic profiling was performed on the cytoplasmic (soluble) fraction and resulted in the identification of 3684 protein groups. Label-free quantification (LFQ) was used to compare inflammatory stimulated with untreated conditions. In LFQ proteomics, the unlabeled peptides are quantified on the MS1 level and the intensities were adjusted to the overall intensity of all quantified peptides. Multi-parameter corrected statistical analysis (FDR = 0.05, S0 = 0.1) revealed 437 and 35 significantly regulated proteins after treatment with TNFα or IL1β, respectively, while 27 protein groups were regulated in both treatments (Figure 1B). Thus, inflammatory stimulation of 12-Z cells by TNFα led to 10-fold higher number of significantly regulated proteins compared to stimulation by IL1β. The most prominent regulations confirmed a successful inflammatory stimulation of the cells by upregulating IL-12 and NF-κB signaling pathways (Figure 1A). Upon TNFα treatment the classical (NFKB1) and alternative (NFKB2, RELB) NF-κB pathways were significantly upregulated while IL1β triggered partly the alternative pathway [51]. Both treatments, however, significantly induced the expression of downstream IL-12 and NF-κB targets (i.e., SOD2, SERPINB2) as exemplified in the heatmap in Figure 1C. Intermediate monocytes are a subpopulation of monocytes characterized by antigen presentation and transendothelial migration [52]. In contrast to classical monocytes, intermediate monocytes feature lower CD14 levels, but increased antigen presentation (HLA), lysozyme, S100A8 and S100A10 as identified by transcriptomic profiling. We found a strikingly similar protein signature corresponding to this intermediate monocyte state for the TNFα-treated, but not the IL1β-treated 12-Z cells (Figure 1D).

### 3.3. TNFα Induces the Expression of Proteins Involved in Neuroangiogenesis in 12-Z Cells

TNFα stimulation showed the upregulation of several proteins involved in neuroangiogenesis. For example, regulated protein groups revealed dorsal root ganglion morphogenesis, sensory neuron axon guidance, neuron projection extension/guidance, semaphorin signaling and positive regulation of sprouting angiogenesis (Figure 2A). ClueGo analysis of significantly regulated proteins further revealed a network corresponding to angiogenesis, including amongst others the VEGF pathway (Figure 2B). Furthermore, the neuropilin receptors NRP1/NRP2 and RPL10 were found to be significantly upregulated [53]. STRING network analysis revealed a strong interconnection of proteins associated with neuroangiogenesis, which were significantly upregulated upon treatment with TNFα, including angiogenesis promoters ICAM1, VCAM1 [54], and ITGA5 [55] (Figure 2C). Furthermore, downstream targets of vascular endothelial growth factor (VEGF) signaling have been found upregulated upon stimulation with TNFα (e.g., BCRA1, ITGAV). A significant upregulation of proteins involved in semaphorin signaling was observed (e.g., OPTN, EPHA4, DHRS3). While none of the described proteins were significantly upregulated by IL1β treatment, they showed a similar trend, which highlighted the differential responses of these endometriotic cells to distinct inflammatory stimulations.

### 3.4. Untargeted Metabolomics Reveals the Upregulation of 7,8-Dihydroneopterin, 7,8-Dihydrobiopterin (BH2) and Normetanephrine in 12-Z Cells

The signature of upregulated proteins involved in neuroangiogenesis and neuropathic pain motivated the investigation of pain-associated signaling molecules on the level of metabolites. For this purpose, an untargeted metabolomics assay was carried out by collecting whole cell lysates and supernatants of control and inflammatory stimulated 12-Z cells. Additionally, the fully supplemented medium was analyzed using the same method to determine the composition of the metabolic background similarly to the eicosanoid analysis. The experiment included a database search based on MS^2^ fragment spectra which resulted in the identification of 29,607 features, from which the software annotated 633 compounds with a match factor ≥80 (Appendix A). After manual review, 63 metabolites were selected and their abundances were quantified on MS^1^ level as area under the curve based on accurate masses and retention time (Appendix A). Multi-parameter corrected statistical analysis (FDR = 0.05, S0 = 0.1) revealed 15 and 4 significantly regulated metabolites for TNFα and IL1β in the whole cell lysates, respectively (Figure 3A). An analogous analysis of the supernatants revealed 3 and 26 significantly regulated metabolites for TNFα and IL1β treatments, respectively (Figure 3B). Strikingly, 7,8-dihydroneopterin, 7,8-dihydrobiopterin (BH2), epinephrine (identity of the molecule verified with external standards) and normetanephrine (annotated based on MS^2^) were detected in endometriotic 12-Z cells (Figure 3 and Figure 4). These metabolites were not detected in the fully supplemented cell culture medium (Appendix A), but only in whole cell extracts and supernatants only in the presence of the 12-Z cells (Figure 4A). The recorded fragmentation spectra matched well the reference spectra from mzcloud database and corroborated the identification of these molecules (Figure 4B). In fact, BH2 and normetanephrine were significantly upregulated in the cellular interior during inflammatory stimulation with TNFα or IL1β, while 7,8-dihydroneopterin was upregulated upon TNFα stimulation only (Figure 3 and Figure 4). The induction of BH2, 7,8-dihydroneopterin and normetanephrine was more pronounced upon activation with TNFα compared to IL1β. Interestingly, the enzymes involved in the biosynthesis of these metabolites remained largely constant upon inflammatory stimulation (e.g., dihydrofolate reductase DHFR or catechol O-methyltransferase COMT), with the exception of sepiapterin reductase (SPR), which was down-regulated by TNFα treatment (Figure 4). Epinephrine and normetanephrine were unexpectedly detected in 12-Z cells, as they were not yet associated with these endometriotic epithelial-like cells.

## 4. Discussion

Although endometriosis is affecting the quality of life of millions of women worldwide, representing a clear unmet medical need, the underlying molecular mechanism of this disease remains largely unknown. As pain sensation is among the most prevalent symptoms, investigating molecular mechanisms responsible for the development of pain may be key to identify useful therapeutic approaches. The interplay among endometriotic cells, macrophages and nerve cells in the ectopic lesions of endometriosis is of special interest for the origin of pain. We performed a multi-omics analysis, including proteomics, metabolomics and eicosanoid analysis, of the epithelial-like 12-Z endometriotic cell line in order to characterize the responses of these cells to inflammatory stimulation and their potential involvement in the development of pain. The 12-Z endometriotic cells were previously characterized as a proliferating and invasive cell line [38]. We found that inflammatory stimulation with TNAα or IL1β did not greatly affect the cell cycle distribution compared to untreated cells. In accordance, the eicosanoid precursors AA, DHA and EPA were efficiently depleted from growth medium irrespective of the inflammatory stimulus and were probably incorporated in the membranes of 12-Z cells. The mono-oxygenated 16– and 18-HETE are cytochrome P450 metabolic products of AA and were released from 12-Z cells. Importantly, 16-HETE is typically generated by exposure of resting neutrophils to AA [56] and represents an endogenous inhibitor of neutrophil activation [57] and thus exhibits anti-inflammatory effects. In our setup, the extent of 16-HETE release was independent of treatment condition.

The proteomic data suggested that the 12-Z endometriotic cells, when stimulated with TNFα, may mimic an intermediate monocytic phenotype, which is generally characterized by transendothelial migration [52], and is actively involved in forming and sustaining a neuroangiogenic microenvironment characteristic for endometriotic lesions. Endometriotic cells were previously shown to exhibit enhanced migratory properties upon exposure to proinflammatory factors [9,10,11]. Moreover, dysfunction in macrophage-mediated phagocytosis of endometrial cells that undergo retrograde transport to the peritoneal cavity is considered an important factor in the development of endometriosis. In fact, this mimicry phenotype of the 12-Z cells may contribute towards the dysregulated immune clearance observed in endometriosis [14,15]. Generally, the 12-Z cells seem more susceptible towards stimulation by TNFα compared to IL1β. Inflammatory stimulation led to the upregulation of proteins involved in neuronal interactions as well as dorsal root ganglion morphogenesis and axon guidance. It is known that the pathology of endometriosis features neuronal interactions and especially neurogenesis [22]. Furthermore, a previous study already showed differential expression of semaphorins and neuropilin receptors correlating to dysmenorrhea [58]. Semaphorins are a group of evolutionarily highly conserved surface or locally secreted nerve repellent factors that can regulate axon migration, axonal growth and guidance [59,60,61]. The potential role of semaphorin 3A and its receptor (NRP1) in the regulation of aberrant sympathetic innervation in peritoneal endometriosis have been previously described [58]. Neuropilin receptors are prominent neurogenesis promoters, which function as axon guidance signaling receptors, as well as angiogenesis activation [31]. Our study shows that stimulation of 12-Z cells with TNFα upregulates the levels of NRP1, NRP2, DPYSL3, OPTN, EPHA4 and DHRS3 proteins suggesting an active involvement of endometriotic epithelial cells in semaphorin signaling. Normally, the process of nervous generation is a conserved feature present during embryonic development [62]. Although proteins like RPL10 and NRP1/2 are mostly associated with embryonic developmental signaling, they have been found significantly overexpressed upon TNFα stimulation in 12-Z in this study and suggest an unrecognized functional plasticity of these cells, which may contribute towards an increased understanding of this pathology.

We further combined the proteome profiling with untargeted metabolomic analysis, investigating whether metabolites of 12-Z cells may be able to contribute to neuronal interaction and signaling. Especially, the capability of the production of 7,8-dihydrobiopterin (BH2), 7,8-dihydroneopterin, epinephrine and normetanephrine by endometriotic epithelial cells was striking. The differential expression of these metabolites was not correlated with the corresponding enzymes in their biosynthetic pathways (Figure 4). 7,8-Dihydroneopterin is an accepted metabolic inflammation marker normally generated by macrophages and has been related to impaired phagocytosis in endometriosis patients [14,63]. Epinephrine is a neurotransmitter secreted by the adrenal medulla. It is required for the vagus-mediated modulation of the nociceptive threshold and acts as inflammatory mediator induced in hyperalgesia [64]. Norepinephrine, the epinephrine precursor, from sympathetic nerve fibers is known to bind the oestradiol ß2 receptor on macrophages, leading to activation of PKA signaling and thus stimulating TNFα mediated inflammation [27]. Norepinephrine is generally involved in inflammation as well as endometriosis pathology [18,65]. This mechanism was previously described as an interaction between nerve cells and macrophages. The deregulation of epinephrine and semaphoring/NRP1 signaling pathways in the nerve cells of endometriosis lesion has been shown to support macrophage polarization [66,67,68]. Our data, however, suggest that epithelial endometriotic cells might themselves be capable of producing these metabolites, subsequently leading to enhanced TNFα secretion by polarized macrophages.

The present model proposes not only a potential influence of endometriosis-associated epithelial cells on macrophages but on nerve cells as well. The significant upregulation of neurogenesis-related proteins demonstrated that the 12-Z cells may be capable of independently modulating neuronal mechanisms. Here again, norepinephrine and its metabolites might play an important role in the activation of pain [69]. It has been shown that epinephrine activates unmyelinated afferents in lesioned nerves [70]. Since endometriotic lesions contain large amounts of unmyelinated nerve fibres [71], there might be a connection between epinephrine secretion and the induction of pain sensation. The detection of epinephrine and normetanephrine in cell lysates and supernatants of IL1β-activated endometriotic cells is unprecedented, since this metabolite is normally produced by adrenal glands. Thus, TNFα and IL1β activation might be important to perpetuate the disease by affecting proteome, metabolome and eicosanoid levels differentially. Finally, the conversion of tetrahydrobiopterin (BH4) into BH2 is involved in the biosynthesis of norepinephrine as the initial hydroxylation step from tyrosine [72]. BH4 application in vivo has been shown to cause heat hypersensitivity and increased pain sensation through TRPV1 [73]. TRPV1 receptor is overexpressed in ectopic endometriosis implants, as well as in dorsal root ganglia of rats with endometriosis [12,74].

## 5. Conclusions

In summary, the data presented in this work highlights the proteomic, metabolomic and eicosanoid alterations upon inflammatory stimulation of the endometriotic epithelial-like cell line 12-Z. Besides the expected activation of inflammatory signaling cascades upon cytokine stimulation, these cells displayed an unexpected protein signature related to neuroangiogenesis which clearly underlined their capability to support neurogenesis in the lesion. Putative novel mediators in endometriosis pathology and pain development were discovered on the protein, metabolite and eicosanoid levels. This study indicates that 12-Z endometriotic cells may mimic an intermediate monocytic phenotype and actively participate in the crosstalk of the macrophage-nervous network within the lesion on the protein and metabolite levels. Thus, inflammatory stimulation of endometriotic cells by TNFα and IL1β seem to play an important role in the perpetuation of the characteristic inflammatory phenotype. They further seem to create factors enhancing the pain sensation through neurogenic inflammation. The actual interaction of endometriotic cells with macrophages and nerve cells requires further investigation but the presented data provided experimental evidence that they might be capable of hijacking immune cell functions in order to support the development and growth of an endometriotic lesion outside the uterine cavity.

## Figures and Tables

**Figure 1 biomolecules-11-01230-f001:**
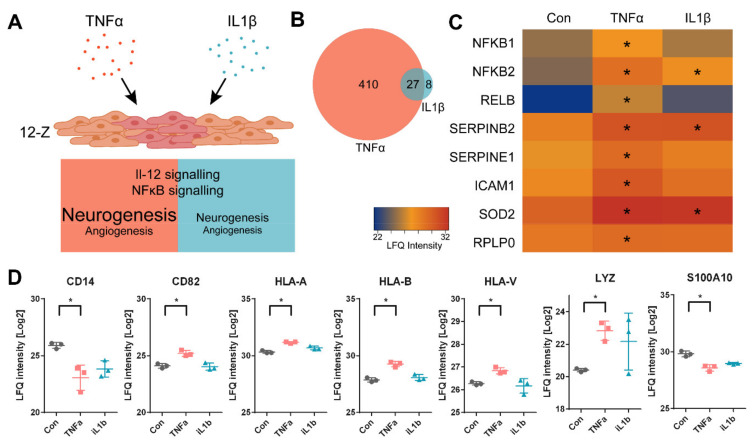
(**A**) Schematic representation of endometriotic 12-Z cells treated with TNFα (red) or IL1β (blue). IL-12 and NF-κB signaling was induced in both treatments while neuroangiogenesis was much more pronounced for TNFα stimulation. (**B**) Venn-Diagram showing the number of significantly regulated proteins (FDR = 0.05, S0 = 0.1) for TNFα (red) and IL1β (blue) compared to the control state. Twenty-seven protein groups were significantly regulated in both treatments. (**C**) Heatmap highlighting proteins involved in NF-κB and Il-12 signaling and downstream targets. (**D**) Protein signature characteristic for the phenotype of intermediate monocytes upon treatment of epithelial-like 12-Z cells with TNFα (red) and not with IL1β (blue). Asterisks (*) show multi-parameter corrected significant regulations of protein abundance (FDR = 0.05, S0 = 0.1) compared to untreated controls (Con).

**Figure 2 biomolecules-11-01230-f002:**
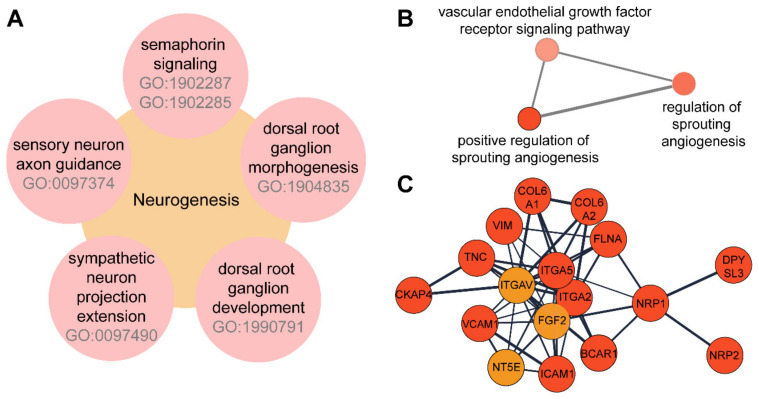
(**A**) Gene Ontology terms for biological processes (GOBP) associated with neurogenesis. (**B**) ClueGo network for angiogenesis. (**C**) STRING network analysis for proteins involved in neurogenesis and angiogenesis. Red indicates multi-parameter corrected significant regulation while yellow-colored proteins show higher expression but are not significant.

**Figure 3 biomolecules-11-01230-f003:**
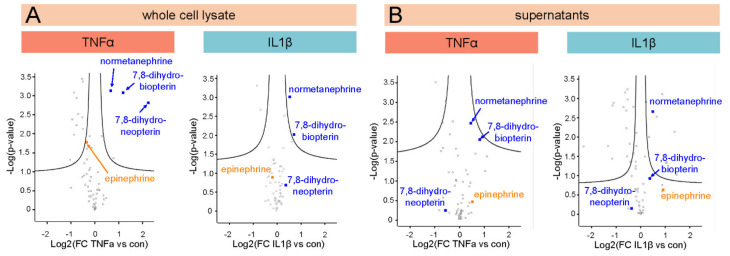
Volcano plots comparing metabolite profiles of control experiments with TNFα and IL1β treatments in whole cell lysates (**A**) and supernatants (**B**). X-axis displays the calculated difference of treatment-control an a log2-scale and y-axis show the -log p-value for each molecule. Metabolites above significance curves represent multi-parameter corrected significantly regulated metabolites (FDR = 0.05, S0 = 0.1).

**Figure 4 biomolecules-11-01230-f004:**
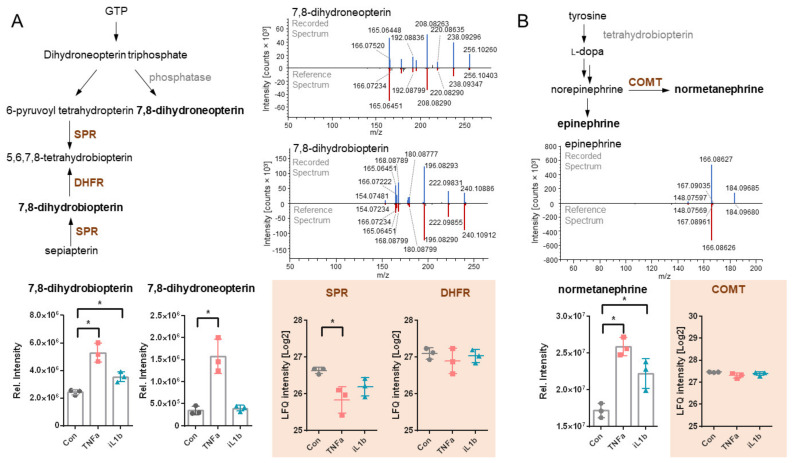
(**A**) The metabolic pathway for the synthesis of 7,8-dihydroneopterin and 7,8-dihydrobiopterin is depicted together with the MS^2^ spectra of the measured metabolites compared to the reference spectrum from the mzcloud database. Intensity values of the metabolite and protein levels (orange background) in control, as well as TNFα– and IL1β-treatment, are given below. (**B**) Biosynthetic pathway of epinephrine derivatives. Normetanephrine is obtained from norepinephrine by the catechol-*O*-methyl transferase (COMT). MS^2^ spectra of normetanephrine compared to the reference spectrum from the mzcloud database. Intensity values of the metabolite and protein levels (orange background) in control, as well as TNFα– and IL1β-treatment, are given below. None of the metabolites were detected in the fully supplemented medium (Appendix A). Asterisks (*) show multi-parameter corrected significant regulations of metabolite intensities compared to untreated controls (FDR = 0.05, S0 = 0.1). The orange shadows distinguish the abundance changes of proteins from those of the metabolites.

## Data Availability

The mass spectrometry proteomics data were deposited in the ProteomeXchange Consortium (http://proteomecentral.proteomexchange.org) via the PRIDE partner repository [39] with the dataset identifier PXD022354 and 10.6019/PXD022354.

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
