# Peer review of "Epithelial Cell Line Derived from Endometriotic Lesion Mimics Macrophage Nervous Mechanism of Pain Generation on Proteome and Metabolome Levels"

_biomolecules, 2021, doi:10.3390/biom11081230_

Round 1

Reviewer 1 Report

The manuscript entitled "Epithelial cell line derived from endometriotic lesion mimics macrophage nervous mechanism of pain generation on proteome and metabolome levels" is of an interesting work. Authors have investigated a role of epithelial-like cells in endometriosis pain development through several interesting parameters. I have some minor comments-

  1. Central message in the abstract is missing.
  2. Some recent findings on endometriosis studies are missing. It would be nice to add some most recent references specially for those sentences where no study is cited i.e. 68-71
  3. Conclusion section should be separated from the discussion. I will recommend to include a figure summering whole study with possible mechanism. 

Author Response

Point 1. Central message in the abstract is missing.

Answer: In our opinion, the central message was stated in the last sentence, which was slightly adapted: “Thus, inflammatory stimulation of endometriotic 12-Z cells led to specific protein and metabolite expression changes suggesting a direct involvement of these epithelial-like cells in endometriosis pain development”.

Point 2. Some recent findings on endometriosis studies are missing. It would be nice to add some most recent references specially for those sentences where no study is cited i.e. 68-71

Answer: We have added two recent studies to the sentences in lines 67–71.

Point 3. Conclusion section should be separated from the discussion. I will recommend to include a figure summering whole study with possible mechanism.

Answer: We have separated the conclusions section from the discussion. An overview figure over the entire workflow is already given in the supplementary information.

Reviewer 2 Report

Gerner et al presents a study by combining proteomics, metabolomics, and eicosanoid analysis to investigate the molecular changes of an endometriotic cell model in response to inflammatory stimulations. The study is well-designed, and the manuscript is well-written. The proteomics results are interesting, and they found a protein signature associated with neuroangiogenesis. This study provides molecular evidence about the involvement of the epithelial-like cells in endometriosis pain development. However, there are several points that the authors need to address:

  1. Introduction part: the authors gave a very comprehensive introduction about endometriosis and inflammatory cytokines, especially TNFα and IL1β, but the literature review about omics studies on endometriosis is lacking. However, this manuscript applied multi-omics technologies (proteomics, metabolomics) to characterize the molecular changes, it makes more sense to have the literature overview in the field, and briefly discuss how this study can fill in the gaps from the multi-omics point of view
  2. It is understandable that the “eicosanoid analysis” in this study is a target method with the detection of 49 eicosanoids and PUFAs with more accurate quantification. Did the authors also detect some of these eicosanoids metabolites in the metabolomics dataset? It is possible to have some overlapping eicosanoids compounds, and if so, did the authors compare the results in the eicosanoids analysis and metabolomics analysis? How were the correlations for the overlapping eicosanoids-metabolites in these two datasets? The authors did not mention it in the manuscript. In addition, the authors did not justify the reasons why it is necessary to include the eicosanoid analysis besides the metabolomics experiment, which is a crucial point. How this eicosanoid analysis adds values to the results?
  3. Metabolomics part: there are 633 compounds with annotation score above 80, the authors mentioned that after manual review, only 63 metabolites were selected. It is surprising that only 63 metabolites were selected for further downstream analysis. It was not described in the manuscript what was the detailed process of this selection based on what criterial? How about the rest of annotated metabolites that were not selected? They should also contain valuable information. The exclusion of these metabolites made this study to lose the opportunity of providing a more comprehensive picture of metabolite changes.
  4. It is unexpected that the differential expression of the metabolites and the corresponding enzymes do not have significant correlations (Figure 4). Did the authors conduct the correlation analysis? What are the correlation values? Did the authors have any hypothesis to explain this? In addition, Figure 4 did not present a good way to indicate this point. What are the reasons why there are orange shadow in Figure 4? It is not described in the figure legend.
  5. The MS/MS spectrum comparison figures in Figure 4 are not very necessary to put as main figures and can be put in the supplementary figures.
  6. The integration of proteomics and metabolomics in this study is relatively weak. Are there any evidence/findings that can be supported by both omics data?

Author Response

Point 1. Introduction part: the authors gave a very comprehensive introduction about endometriosis and inflammatory cytokines, especially TNFα and IL1β, but the literature review about omics studies on endometriosis is lacking. However, this manuscript applied multi-omics technologies (proteomics, metabolomics) to characterize the molecular changes, it makes more sense to have the literature overview in the field, and briefly discuss how this study can fill in the gaps from the multi-omics point of view

Answer: We thank the reviewer for emphasizing this point. We have now included a paragraph about omics studies in endometriosis and added appropriate references.

Point 2. It is understandable that the “eicosanoid analysis” in this study is a target method with the detection of 49 eicosanoids and PUFAs with more accurate quantification. Did the authors also detect some of these eicosanoids metabolites in the metabolomics dataset? It is possible to have some overlapping eicosanoids compounds, and if so, did the authors compare the results in the eicosanoids analysis and metabolomics analysis? How were the correlations for the overlapping eicosanoids-metabolites in these two datasets? The authors did not mention it in the manuscript. In addition, the authors did not justify the reasons why it is necessary to include the eicosanoid analysis besides the metabolomics experiment, which is a crucial point. How this eicosanoid analysis adds values to the results?

Answer: We did not find overlapping eicosanoids in the eicosadomic and metabolomic data sets. Eicosanoids are lowly abundant signaling lipids, which require an enrichment step during sample preparation. This was not performed during the sample workup for the metabolomic analysis. Moreover, the chromatographic conditions in both setups are very different and discriminate against lipidic species in the metabolomic analysis.

Eicosanoids are well-known signaling lipids in the acute phase of inflammation. Since we investigate the effect of inflammatory activation on endothelial-like endometriotic cells, we believe it is interesting to investigate and report changes in eicosanoid/oxylipin signaling in this context.

Point 3. Metabolomics part: there are 633 compounds with annotation score above 80, the authors mentioned that after manual review, only 63 metabolites were selected. It is surprising that only 63 metabolites were selected for further downstream analysis. It was not described in the manuscript what was the detailed process of this selection based on what criterial? How about the rest of annotated metabolites that were not selected? They should also contain valuable information. The exclusion of these metabolites made this study to lose the opportunity of providing a more comprehensive picture of metabolite changes.

Answer: The 63 metabolites were selected due to their connection to the disease model. Additionally, these metabolites were verified by external standards and thus represent a collection of informative and confidently identified metabolites that are relevant for this in vitro model. The screening process actually included 633 compounds. If we had found other apparent regulatory events of potential interest, we would have verified and validated such candidates as we did in case of the present findings.

Point 4. It is unexpected that the differential expression of the metabolites and the corresponding enzymes do not have significant correlations (Figure 4). Did the authors conduct the correlation analysis? What are the correlation values? Did the authors have any hypothesis to explain this? In addition, Figure 4 did not present a good way to indicate this point. What are the reasons why there are orange shadow in Figure 4? It is not described in the figure legend.

Answer: We disagree with the reviewer. It is actually expected that the abundance of proteins and metabolites hardly correlate because changes in enzyme concentration do not correlate necessarily with enzymatic activity and turnover. It would be required to perform flux analyses to connect metabolite concentrations to protein expression, which would go beyond the scope of this study.

The orange shadows were intended as a visual aid to distinguish between the metabolites and proteins. The figure legend was updated.

Point 5. The MS/MS spectrum comparison figures in Figure 4 are not very necessary to put as main figures and can be put in the supplementary figures.

Answer: We have included the MS/MS spectrum comparison for those readers who are not particularly familiar with metabolomic analyses based on mass spectrometry. Therefore, we would prefer to keep them in Figure 4.

Point 6. The integration of proteomics and metabolomics in this study is relatively weak. Are there any evidence/findings that can be supported by both omics data?

Answer: The inflammatory stimulation of endometriotic 12-Z cells led to specific protein and metabolite expression changes that independently suggested their involvement in pain development. We consider this a clear benefit in the multi-omics approach in this study. Most importantly, the unexpected finding of pain-associated processes was independently confirmed on both proteome and metabolome levels, which served to validate each other in a mutual fashion.